# A Distributed Approach to Speaker Count Problem in an Open-Set Scenario by Clustering Pitch Features

**Sakshi Pandey** [†] [iD] **and Amit Banerjee** *,[†] [iD]

Department of Computer Science, South Asian University, New Delhi 110021, India; cs.sakshi.04@gmail.com
* Correspondence: amit@cs.sau.ac.in
† These authors contributed equally to this work.

**Abstract:** Counting the number of speakers in an audio sample can lead to innovative applications, such as a real-time ranking system. Researchers have studied advanced machine learning approaches for solving the speaker count problem. However, these solutions are not efficient in real-time environments, as it requires pre-processing of a finite set of data samples. Another approach for solving the problem is via unsupervised learning or by using audio processing techniques. The research in this category is limited and does not consider the large-scale open set environment. In this paper, we propose a distributed clustering approach to address the speaker count problem. The separability of the speaker is computed using statistical pitch parameters. The proposed solution uses multiple microphones available in smartphones in a large geographical area to capture and extract statistical pitch features from the audio samples. These features are shared between the nodes to estimate the number of speakers in the neighborhood. One of the major challenges is to reduce the error count that arises due to the proximity of the users and multiple microphones. We evaluate the algorithm's performance using real smartphones in a multi-group arrangement by capturing parallel conversations between the users in both indoor and outdoor scenarios. The average error count distance is 1.667 in a multi-group scenario. The average error count distances in indoor environments are 16% which is better than in the outdoor environment.

**Keywords:** speaker count; distributed architecture; prosodic parameters; statistical parameters; node clustering; feature clustering

## 1. Introduction

Advancements in smart devices have surged the demand for applications that can serve customized user experiences in real-time, such as finding nearby restaurants with a high rating. The social interaction among humans can be analyzed to extract various attributes like the content and context of communication. Determining the number of speakers in a conversation is one such attribute, commonly referred to as the speaker count problem. It can be useful for applications such as real-time ranking systems [1]. The popular ranking systems are based on user feedback collected in text format [2]. In general, the ranking algorithms statistically quantify the user's feedback to rank the popularity/usefulness of a product or an object. These ranking systems are useful for determining the popularity of a restaurant or movie. However, offline rankings systems can be faked for creating false publicity of a product [3]. A real-time ranking system can be a solution for the same, which is based on the assumption that a place/object's popularity is directly related to the number of distinct users present in its proximity.

Researchers have studied the speaker count problem by synthesizing the audio [4,5], video [6] and images [7] of an instance. In general, most of these techniques use supervised learning approaches to determine the number of speakers in a conversation and rely on external servers to maintain a predefined data set for learning and matching. For example, in [8], authors assume a closed scenario and use a supervised learning approach for single-channel speaker count estimation. Similarly, [9] propose an algorithm to estimate the

number of sound sources in a realistic environment. Apart from supervised learning approaches, researchers have considered un-supervised algorithms to solve the speaker count problem by learning the audio signal's statistical properties to estimate the number of speakers [4,5]. However, there are several existing challenges to the speaker count problem. For example, the error in determining the total speaker count increases if multiple microphones are used for collecting the audio samples from a large geographic area [5]. That is, the microphones can collect multiple data samples of the same speaker, which is difficult to parse by simple additive theory.

This paper aims to study the speaker count problem in an open-set scenario using multiple microphones, as shown in Figure 1. More specifically, we investigate a quasi-distributed architecture [10] to address the speaker count problem in a large geographic area using multiple microphones. In the architecture, a group of SDs can exchange the locally generated features to generate the final speaker count. The paper uses the concept of controllers (or beacon nodes) for enhancing the scalability of the system and reduce the network load when the coverage area increases in real-time scenarios. The paper aims to study a distributed solution for the speaker count problem that can be implemented on smart devices for various real-time applications, such as ranking the popularity of a restaurant or movie theater. The proposed algorithm uses prosodic parameters for the separability of human voices [11]. The advantage of using the prosodic parameter is that it is less vulnerable to channel distortion and noise than the acoustical parameters [12,13].

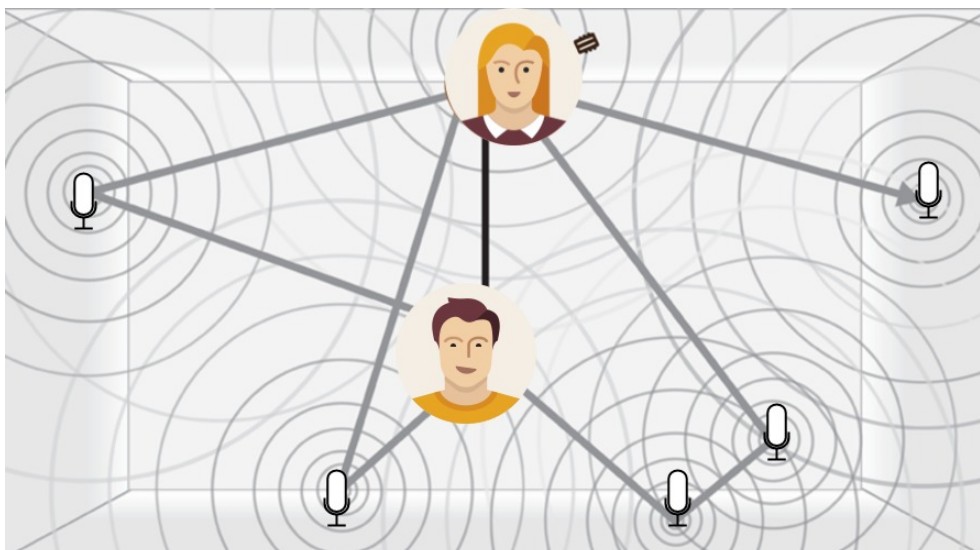

**Figure 1.** Speaker's proximity to multiple microphones.

Crowd++ [5] is closely related to our work. In Crowd++, MFFC is combined with the pitch parameter to count the number of speakers. Before this, Agneessens et al. in [4] used the pitch estimation algorithm to differentiate a single speaker recording and two speaker recordings on the mobile devices. However, the primary differences between Crowd++ and the proposed methodology are as follows:

- Crowd++ combines Pitch with MFCC (a general-purpose feature in speech processing) for estimating the speaker count. Whereas in the proposed algorithm, we extract features only from the Pitch to estimate the speaker count. Thus, minimizing the computational overheads in terms of real-time factors.
- Crowd++ states it is an entirely distributed approach as it does not have any infrastructural requirements, at the same time, it simply adds the results of the multiple microphones (in-case of multi-group scenarios) to estimate the speaker count, overlooking the proximity of a speaker to multiple microphones (as shown in the Figure 1), which may result to over-count. As a solution to the problem, the proposed distributed

approach performs a periodic exchange of extracted statistical features, which results in a more accurate approximation of speaker count.

- In Crowd++, each SD runs the application individually, which may result in variable results due to various challenges like the phone's location (in or out pocket), SD's hardware. While in the proposed distributed approach, all the participating SD's will generate the same result.
- The proposed distributed approach provides a setup enhancing the system's scalability, while there is no such discussion about it in Crowd++.

Figure 2 shows the architecture of the proposed distributive speaker count system, containing smart devices (SDs), controllers (CTLs), and users. The SDs can be any device with a microphone and processing capability, such as smartphones or smartwatches. We assume that the SDs are distributed randomly in a given geographical area and grouped into multiple clusters to improve the system's scalability. The SDs are responsible for collecting data, processing, and sharing information with other SDs within the cluster. The CTLs are used for exchanging the inter-cluster information and for sharing the speaker-count information with the users.

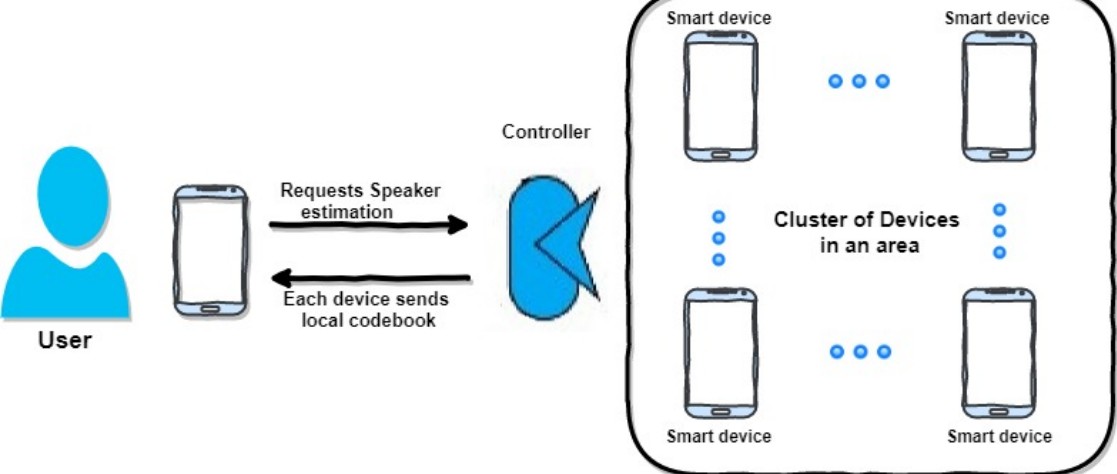

**Figure 2.** System architecture.

The three modules of the proposed architecture are data acquisition, feature extraction, and feature clustering. The data acquisition module captures the audio signal and calculates the instantaneous pitch of a signal. The feature extraction module uses the pre-processed pitch information to evaluate various statistical parameters, such as mean, standard deviation, skewness, and kurtosis for each block. Finally, feature clustering is used to group similar pitch features from multiple SDs, reducing the redundancy in data and speaker count error. The challenges of the proposed methodology are as follows:

- Location of phone: The position of the phone (e.g., inside a pocket or a bag) affects the sensitivity of the microphone, which may result in an over-count or under-count of speakers.
- Background Noise: In real-world scenarios, other sources of interference are voice generated by TV or radio equipment which can cause over-count.
- Real-time speaker count: Speaker count in a real-time scenario is a challenging task, as the cluster identification time depends on the distances between the feature vector of an unknown speaker and dataset.
- Proximity to Microphones: As we are considering multiple microphones in the region of interest, the proximity of a speaker to multiple phones should not affect the overall speaker count.
- Open-set speaker count: In a large-scale real-time environment, the system should not consider any prior knowledge of the speakers in the audio sample.

- Scalability: The system must handle dynamic and crowded environments.

For the experimental setup, we use android smartphones for audio recording and pitch feature extraction. The pitch features are shared with other SD's for aggregation and reduction of the error count. In the experiments, we empirically calculate the threshold for inter-and intra-cluster distances ($\theta$) by extracting the pitch features, using the audio recordings of 2~7 persons. The $\theta$-value is used for evaluating the distributed speaker count in both indoor and outdoor environments. For this, we collect the data samples of 30 min using the smartphones of 10 friends for a week. For indoor audio samples, we consider our lab and the university canteen during the peak hours of lunchtime for maximum background noise. We also evaluate the architecture in a multi-group environment where parallel conversations are taking place simultaneously, and multiple phones are collecting the audio samples.

The organization of the article is as follows. In Section 2, we discuss various related works of the speaker count problem. Section 3 provides a system overview of the proposed methodology. Section 4 discusses the experiment results and evaluation. Section 5 discusses about the approach. Finally, we conclude our work in Section 6.

## 2. Related Works

Researchers have studied the speaker count problem for audio [4,5], video [6] and images [7]. In [7], the authors use two cameras and two microphones to perceive the scene. A multimodal Gaussian mixture model (mGMM) fuses the information extracted from the auditory and visual sensors and detects the most probable audio-visual object, e.g., a person emitting sound in 3D space. Audio and visual cues used in [6] apply a likelihood-based approach for speaker identification using pure speech data and apply techniques such as face detection/recognition and mouth tracking for talking face recognition using pure visual data. Additionally, using the audio signal features are extracted for analysis purposes.

In the following discussion, we focus only on the audio samples, as we can extract the results in a real-time environment via lightweight computational devices like mobile phones [4,5]. Generally, a speaker recognition system differentiates the speakers in an audio sample while ignoring the content. For the same purpose, we need to extract certain characteristics or features from a speech signal for identifying an individual. Broadly, the proposed solutions to the speaker count problem can be categorized into supervised [8,9] and un-supervised [14] approaches. The supervised algorithms generally use prior user information for the classification of the speaker. In [9], authors propose a unifying probabilistic paradigm, using deep neural network architectures to infer output posterior distribution. Recently, researchers have used deep learning [15], and neural-networks [16] for the speaker count problem. However, environmental changes in the above algorithms can lead to poor real-time performance.

In comparison, un-supervised solutions to the speaker count problem generally use statistical methods for estimating the number of speakers in an open-set scenario as no prior knowledge of the speaker is available. The characteristics of a speaker's voice are reflected by the channel and glottal features [17]. Research suggests that the speaker-recognition systems depend on the spectral features extracted from very short time segments (sec frame) of a speech, formally known as MFCC [12]. The MFCC based approach is highly successful in a noise-free environment, but the performance degrades significantly with variability in the channel. The other approach used is the long-range features such as lexical, prosodic, and discourse-related habits [12]. Prosodic features are the rhythmic and intonational properties in a speech, for example, the voice fundamental frequency (F0), F0 gradient (pitch), intensity (energy), and duration. These are relatively simple in structures and are considered effective in speech recognition [18].

Next, we discuss the works that use audio processing for the speaker count problem. Ofoegbu et al. in [19] study the problem of finding the number of speakers in an audio sample of four speakers using a generalized residual radio algorithm. The approach is

based on machine learning algorithms built on supervised training techniques. Similarly, in Speaker diarization [20], authors essentially determine "who-spoke-when" in an audio recording using computationally expensive models (Gaussian mixture model—GMM, HMM) and algorithms (BIC, MCMC). Crowd++, as discussed above, uses MFCC and the pitch parameter to count the number of speakers. The authors use unsupervised machine learning analysis on audio segments captured by smartphones. Agneessens et al. [4] also discuss the speaker count problem using pitch estimation algorithm to differentiate between single and two speaker recordings.

In comparison to the above approaches, in this paper, we propose a distributed clustering architecture to enhance the scalability of the system. Besides, we use the statistical evaluation of the prosodic parameter (pitch) to capture the speaking behavior of the speakers. The main objective is to study the robustness of the algorithm in a real-time environment.

## 3. Distributed Speaker Count

### 3.1. System Architecture

Figure 2 shows the system architecture of the proposed methodology consisting of three basic entities: smart devices (SDs), controller, and users. The SDs are the devices with a microphone and processing capability, such as smartphones or smartwatches, and are responsible for data collection and processing. Node clustering techniques can be used on the SDs for improving the scalability of the system [21,22]. Researchers have proposed various clustering techniques in wireless ad hoc and sensor networks, such as using dominating set and beacon nodes [23–25]. Similarly, in 2G or 3G cellular networks, coverage areas of the base stations (or cells) can be considered as clusters covering a geographic area.

In the proposed system, the controller nodes are used for clustering the network. For simplicity in the implementation, we manually select a subset of SDs to serve as the controller nodes, which periodically sends beacon messages to partition the SDs into mutually disjoint clusters. The number of clusters in the network is equal to the number of controllers present in the system. On receiving the beacon messages, the SDs select a controller node based on its proximity and sends a "join" message to the corresponding controller. After receiving all join messages from the SDs, the controller sends the member information back to all SDs. Thus, the member information is known to all nodes present within the cluster. A cluster's controller is generally responsible for maintaining various intra- and inter-cluster information [26]. Apart from clustering, the controller nodes act as an interface to provide the speaker count information for the cluster.

Finally, the "user" in Figure 2, are the nodes (or applications) interested in the speaker count information. For this, a user sends a "speakerinfo" message to the corresponding controller. On receiving this request, the controller can send the speaker count information back to the user. The controller can use periodically generate the speaker count information and cache the result to reduce the network's communication overhead. We can apply hierarchical clustering techniques [22] to improve the scalability of the network and reduce the network load. More details on various distributed node clustering techniques can be found in [27,28].

### 3.2. System Modules

In this section, we use Figure 3 to provide a system overview and explaining the three major components executed by the SDs in the proposed system, namely, the data acquisition module, feature extraction module, and the clustering module. Given below are the details of each module and the general structure of the proposed algorithm.

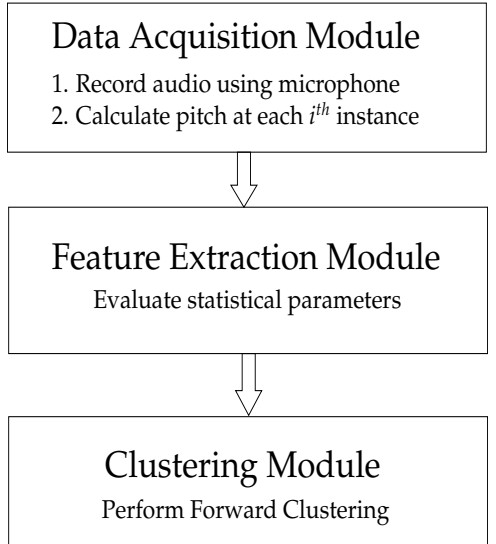

**Figure 3.** System modules of the proposed distributed speaker count algorithm.

### 3.2.1. Data Acquisition Module

The data acquisition module performs two basic tasks, i.e., data collection and data pre-processing. Let '$y$' be a time-domain real-time audio signal recorded by the microphone present on a smart device (SD). The SD periodically calculates the instantaneous pitch of the audio sample. The pitch estimation uses the YIN algorithm [29], which is a time-domain pitch calculation algorithm based on autocorrelation. The YIN algorithm is energy-efficient, making it suitable for the resource-constraint SDs [30]. The physiological feature of the speaker's vocal chord limits the variation of the pitch. The typical pitch interval of a human voice is 50 to 450 Hz [31]. Voice activity detection uses the pitch at an instance. The YIN algorithm converts the input speech signal '$y$' to an instantaneous pitch P, as shown in Figure 4. Further, the pre-processed data P estimates the statistical pitch feature '$y$'.

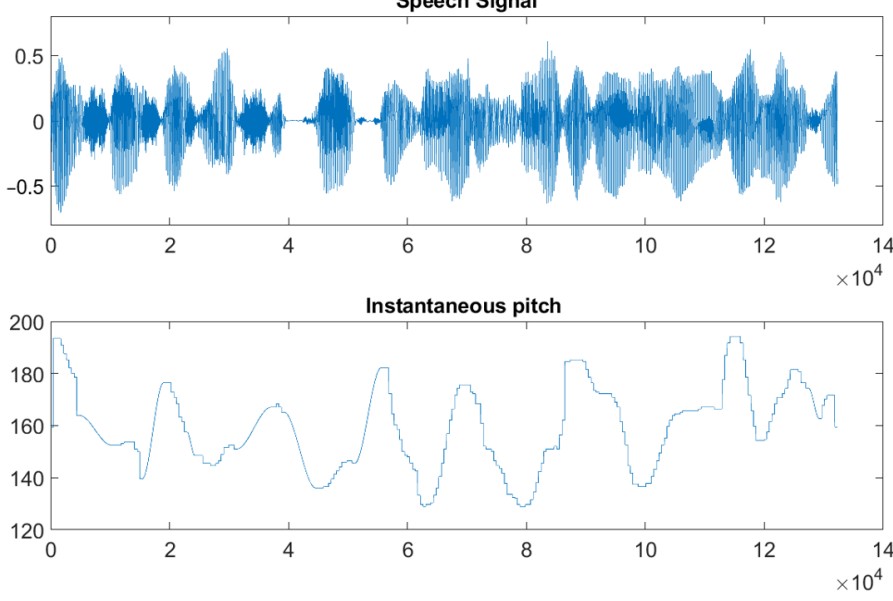

**Figure 4.** Speech Signal to instantaneous pitch.

### 3.2.2. Feature Extraction Module

This module handles feature extraction by using the pre-processed information received from the data acquisition module. Periodic evaluation of statistical pitch features

takes place using the pitch $P$ at each instance. In this step, instantaneous pitch $P$ is converted into $N$ equal-sized blocks $P = \{P_1, P_2, P_3, \cdots, P_N\}$, where each block stores the evaluated statistical pitch features. In our implementation, we consider the block size of 3 s, which in general is the turn-taking pattern in everyday communication [30]. For each $i$th block, we define a feature vector $P_i = \{\mu, \sigma, \gamma_1, \gamma_2\}$, $\mu$ is the mean pitch value, $\sigma$ is the standard deviation of the pitch value, $\gamma_1$ is the skewness and $\gamma_2$ is the kurtosis for each block.

### 3.2.3. Feature Clustering Module

Finally, the SDs use the extracted pitch features to generate the local codebook. Forward clustering is used to group a similar pitch feature. The idea behind this is the temporal coherence in the speech, i.e., there is a high probability that the consecutive blocks belong to the same speaker. Thus, forward clustering performs a pairwise comparison of the consecutive statistical pitch features by calculating the Euclidean distance between each pair. That is, it computes the Euclidean distance ($d_{ed}$) of blocks $P_1$ and $P_2$, and if $d_{ed}(P_1, P_2) < \theta$ (where $\theta$ is the threshold distance), we merge the two blocks into a single block $P_1$, as shown in Figure 5. Next, it calculates the Euclidean distance of $P_1$ with $P_3$ and merges them into a single block, if $d_{ed}(P_1, P_3) < \theta$, otherwise compare $P_3$ to $P_4$ and so on. The process of merging the blocks generates a block-specific VQ codebook, described by the LBG algorithm [32]. The resulting codeword is the centroid of all the statistical pitch feature vectors that lie within the same range of similarity. SDs periodically exchange the codebook with each other.

$$P_1 = \frac{1}{2}(P_1 + P_2), where\ P_i = \{\mu, \sigma, \gamma_1, \gamma_2\} \tag{1}$$

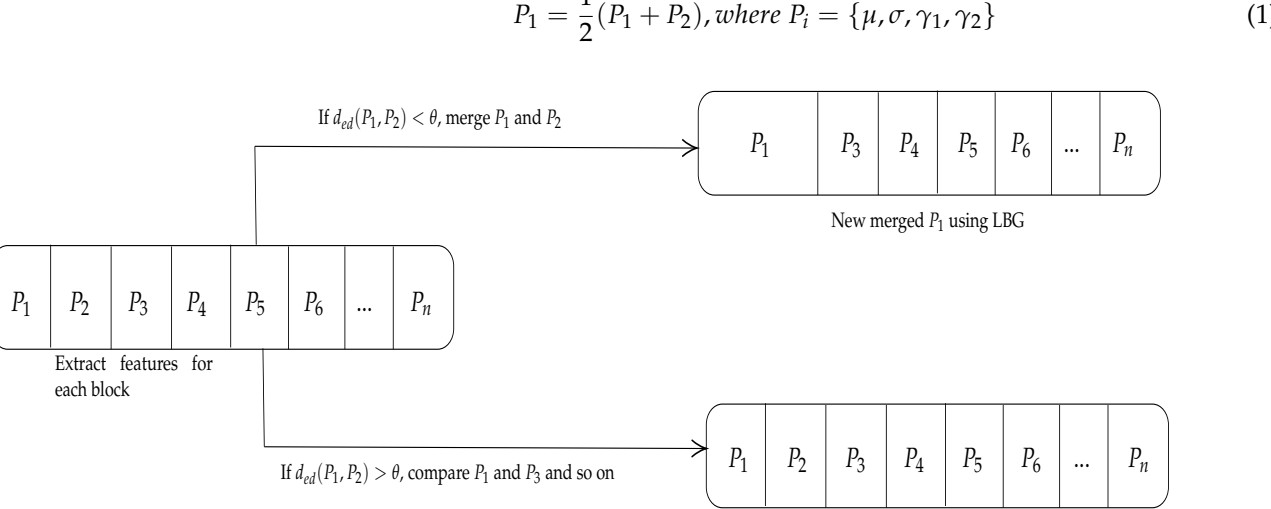

**Figure 5.** Forward clustering.

### *3.3. Proposed Algorithm*

As mentioned above, this paper's focus is to determine the number of speakers by processing the audio samples collected by multiple microphones. Algorithm 1 is the pseudocode used for the proposed distributed speaker count problem. The algorithm shows the responsibilities of the SDs and CTLs. The SDs use the function *SD_LocalCodeBook()* to generate its local codebook. For this, SDs with a microphone capture the audio signal and processes it to extract the instantaneous pitch and determines the speakers' statistical features. Finally, the SDs perform feature clustering over the pitch feature vector to generate VQ (Vector quantization) codebook by merging the similar features [33], discussed in Algorithm 2.

The codebook is shared with other SDs within the cluster using *ShareLocalCodebook()* to reduce the intra-cluster speaker count error, which occurs due to the proximity of the speakers and multiple microphones placed within a cluster. For example, in Figure 1,

the user's pitch features extracted by the microphones can vary depending upon various factors, such as its distance, strength of the microphones, and obstacles between the two. To resolve this, the SDs share their local codebook within the cluster and merge them using *MergeCodebook()*, (Algorithm 1), which concatenates multiple feature vectors, i.e., $P' = \{P_1' \cup P_2' \cup \ldots\}$ and calls *FeatureClustering(P)* (Algorithm 2) to remove the redundancies.

The SDs present along the border of two or more clusters can capture audio samples from multiple clusters and increase the total speaker count. We refer to it as an inter-cluster speaker count error. To resolve this, we take the help of the CTLs. Notice that a CTL of a cluster can generate the codebook similar to the SDs present within its cluster. To reduce the inter-cluster speaker count error, each CTL periodically shares its codebook with the CTLs of the neighboring clusters using *CTL_ShareCodeBook()* in Algorithm 1. The CTLs use *GenGlobalCodebook()* to merge the received codebooks and generate the final speaker count.

---

**Algorithm 1:** System Modules of the Proposed Distributed Speaker Count Algorithm.

---

**1 Function** `DataAcquisition(`$y$`)`
  // Parameter $y$ is the speech signal
**2**   **while** *end of(y)* **do**
**3**    **for** *each instant i* **do**
      // calculate the instantaneous pitch $p_i = \{p_1, p_2, p_3, \cdots, p_N\}$
**4**     **if** *(50 $\leq$ pitch(i) $\leq$ 140)* **then**
**5**      $p \leftarrow$ Store the pitch value $p_i$
**6**   **return** $p$

**1 Function** `FeatureExtraction(`$p$`)`
  // Parameter $p$ is an array obtained from DataAcquisition module
**2**   **for** *each $p_i \in p$* **do**
      // for fixed size block($P_i$)
**3**    Compute statistical pitch feature vector $P_i = \{\mu, \sigma, \gamma_1, \gamma_2\}$
**4**    $P \leftarrow$ Store the feature vector $P_i$
**5**   **return** $P$

**1 Function** `FeatureClustering(`$P$`)`
  // Parameter $P = \{P_1, P_2, \cdots\}$ is the feature vector from
    FeatureExtraction module
**2**   **for** *each pair $(P_i, P_j) \in P$* **do**
**3**    **if** *($d_{ed}(P_i, P_j) < \theta$)* **then**
      // $\theta$ is the threshold distance
**4**     Merge the blocks.
**5**     $CB \leftarrow$ Generating VQ codebook using LBG algorithm
**6**   **return** $CB$

---

---

**Algorithm 2:** Distributed Speaker Count Algorithm.

```
  // Functions executed by the SDs
1 Function SD_LocalCodeBook()
2 │   ReadAudioSample()                                // collect audio sample
3 │   DataAcquisition()                          // instantenious pitch calculation
4 │   FeatureExtraction()              // extarct the feature from the audio sample
5 │   FeatureClustering()            // feature Clustering and generate codebook
6 │   ShareLocalCodebook() // share local codebook with neighbors in the cluster

1 Function SD_OnReceivingCodeBook()
2 │   MergeCodebook()     // Merge the codebooks received from neighboring SDs
3 │   ShareMergedCodebook()     // periodically share merged codebook within the
  │     cluster

  // Responsibilities of the Controllers (CTLs)
  // CTLs shares the codebook to compute the speaker count of a region
1 Function CTL_ShareCodeBook()
2 │   ShareCodebook()       // Periodically share codebook with other controllers

1 Function CTL_ShareCodeBook()
2 │   GenGlobalCodebook()              // Merge codebooks received from other CTLs
3 │   SC_i ← Compute speaker count     // Compute and store the speaker count at
  │     instance i
```

---

## 4. Experimental Results

In the following, we evaluated the performance of the proposed distributed architecture for both single- and multi-group scenarios. A single microphone may not cover the sizeable area for various reasons, such as attenuation of the audio signal or range limitation of the microphones. For this reason, we considered several microphones to collect data, which were placed randomly without any specific geometrical arrangement. We used smartphones for data collection and processing. We developed an application using Java for the Android platform and installed it on smartphones for recording the raw audio at an 8 kHz frequency, 16-bit pulse-code modulation(PCM).

In our experiments, we took the help of 10 friends to collect data for a week for 30 min in both indoor and outdoor environments. The data samples were collected for 30 min using the smartphones of 10 friends for a week. For indoor audio samples, we considered our lab and the university canteen during the peak hours of lunchtime for maximum background noise. For the outdoor data samples, the data were collected from the university playground to study the effect of the background noise. The results below are the average of the above-collected data for the single and multi-group scenarios. For the single-group case, only one smartphone was used for data collection in our lab's indoor environment. The performance of the multi-group scenario was an average of all data collected for the three environments. We manually recorded the number of speakers in each case for evaluating the performance of the proposed algorithm.

To evaluate the accuracy of the proposed algorithm, we calculated the Error Count Distance (ECD), which is the difference in the number of speakers obtained from the proposed method $\hat{k}$ and the actual number of speakers $\hat{k}$, i.e., $|\hat{k} - k|$. We use the absolute value to avoid negative distances. The average error count shows the accuracy of the algorithm. In the following, we first discuss the calculation of threshold ($\theta$) (i.e., the separability of voice samples) for voice samples. We use the threshold for evaluating the number of users in single and multiple group scenarios.

### 4.1. Evaluation of the Threshold ($\theta$)

To calculate the inter-and intra-cluster threshold ($\theta$), we collected the audio recordings of 2∼7 individuals, both male and female, in the age group of 23∼27 years. We collect 10 audio samples of about 90 seconds for each individual, assuming that there was no

overlap in the voice segments. The statistical pitch features of $N$ blocks, i.e., $(\mu, \sigma, \gamma_1, \gamma_2)$ as discussed in Section 3.2.2, formed a vector in the 4-dimensional vector space. Figure 6 shows the cluster formed by vectors for 1 and 2 speakers. The Euclidean distance between two points $X$ and $Y$ in the vector space is calculated using $d(X, Y) = \sqrt{\sum_{i=1}^{n}(x_i - y_i)^2}$, where $n$ is the number of statistical parameters. Equation (2) calculates the similarity between the points in the vector space, where $R$ is the range of similarity, $q$ is the centroid, and $r$ is the separability distance (defined as $\theta$) of the cluster. In the paper, we refer to $\theta$ as the threshold for determining the intra-cluster distances for the speaker count estimation.

$$R(q, r) = \{s \in X, d(s, q) \leq r\} \tag{2}$$

Figure 7 shows the inter and intra-cluster distances for 1~7 speakers. It is evident from the figure that the intra-cluster distance was almost the same for all cases. However, the inter-cluster distance for a male-female voice sample was comparatively more than that of the same gender. So, the challenge was to distinguish between the same gender voices. Table 1 shows the centroids for three speakers, and Table 2 shows the min, max, and average inter-cluster distances for an increasing number of speakers. We use Table 2 and Figure 7 to set the threshold ($\theta$) of the proposed methodology as 13, which was greater than the maximum intra-cluster distance of the same gender.

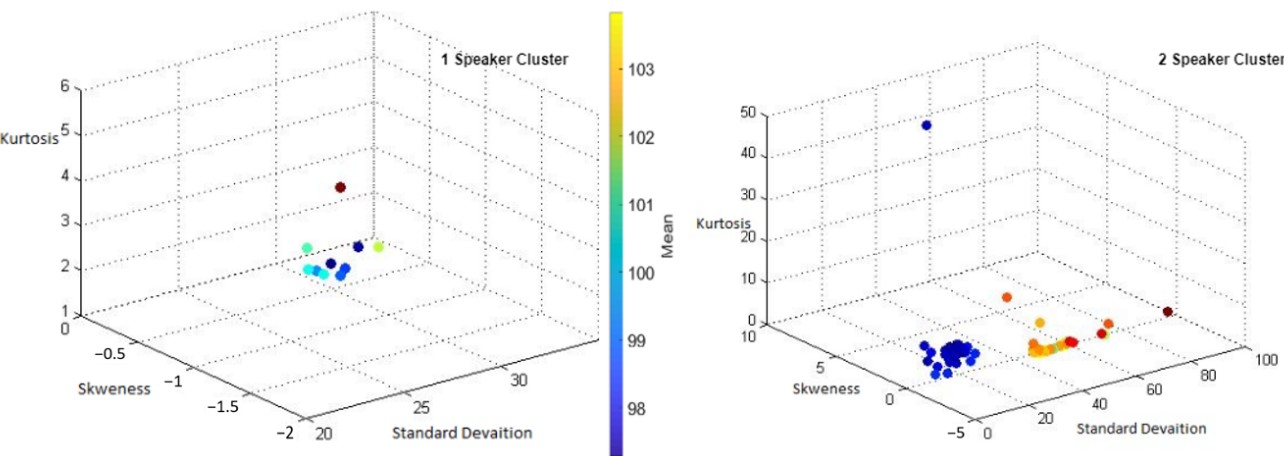

**Figure 6.** Cluster formed by 1 and 2 speakers.

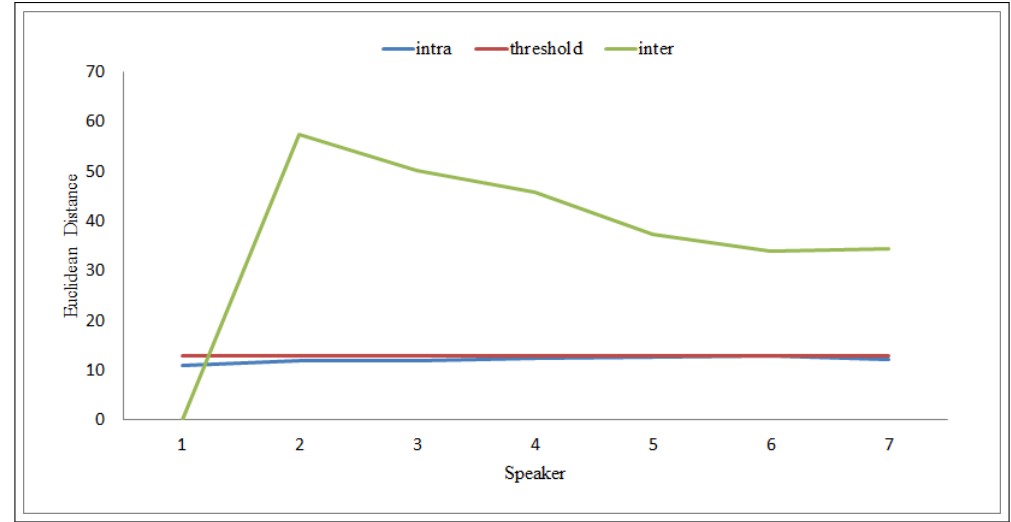

**Figure 7.** Inter-, intra- and threshold Euclidean distances.

**Table 1.** Threshold($\theta$) distances for 3 speakers.

| Attributes | Clusters | | |
|---|---|---|---|
| | **1** | **2** | **3** |
| Size | 25 | 33 | 32 |
| $\mu$ | 118.4341 | 177.4756 | 118.334 |
| $\sigma$ | 29.3108 | 51.5388 | 14.2691 |
| $\gamma_1$ | $-0.8291$ | $-0.9232$ | $-0.1311$ |
| $\gamma_2$ | 3.4473 | 3.4345 | 6.9116 |

**Table 2.** Inter-cluster distances.

| Speaker | Inter Cluster | | |
|---|---|---|---|
| | **Average** | **Min** | **Max** |
| 4 | 45.9259 | 26.44 | 69.97 |
| 5 | 37.489 | 28.21 | 56.16 |
| 6 | 34.08 | 28.69 | 58.75 |
| 7 | 34.444 | 28.45 | 56.12 |

*4.2. Performance in Single Group Scenario*

In this experiment, we used a single microphone to record the conversation of multiple speakers. Above we have already discussed the experimental setup. Depending on the type of conversation, the two possible cases in a single group scenario were (a) speakers taking turns in the communication or (b) speaking simultaneously, resulting in an overlapping speech sample. As suggested in [34], if the frame size was small in a general public conversation, then the possibility of finding an overlapping voice reduced. In the following, we chose a frame-size of 3 s to reduce the chances of error in the speaker count due to overlapping segments. Although we could reduce the frame size to improve the accuracy, it may lead to oversampling and increase the complexity of the algorithm.

Figure 8 shows the average error count distance for 1~10 speakers in a closed lab environment. The Figure shows that there was a linear increase in the error count distance with an increasing number of speakers. The average error count distance of all cases was 0.47. In Figure 8, there was a sudden increase in the error count, as in the case of 3 speakers. A primary reason for this can be the presence of the same gender in the voice sample. If we had two male or female speakers in an audio sample, there was a misclassification error of 10% in the clustering (Table 1). Moreover, the errors got prominent with an increasing number of speakers, as reflected in the data. The prosodic parameter pitch varied with emotion and can affect the overall speaker count.

*4.3. Performance in Multiple Groups Scenario*

In the following, we considered a scenario where multiple groups of people were near each other, and each group was using a smartphone to record the audio. As discussed above, the data were collected for 10 speakers for a duration of 30 min in a closed and crowded indoor location and an outdoor environment. We tested our algorithm in two and three group scenarios. The experimental results discussed below are the average of 20 voice samples.

4.3.1. Performance of Two Group Scenario

In this, there were five participants in each group. The speaker count evaluated by each SD is shown in Figure 9. The figure shows that the estimated speaker count in the indoor condition was better than the outdoor environment. Figure 10 shows the average speaker count in all three cases. The figure compares the proposed distributed

approach and the additive approach of the speaker count estimated by each node (usually followed approach) in the region of interest. The result showed that the proposed distributed approach was better for all three cases, with an average error count distance of 0.33. The distributive architecture helped to solve the problem of multiple microphones recording parallel conversations simultaneously.

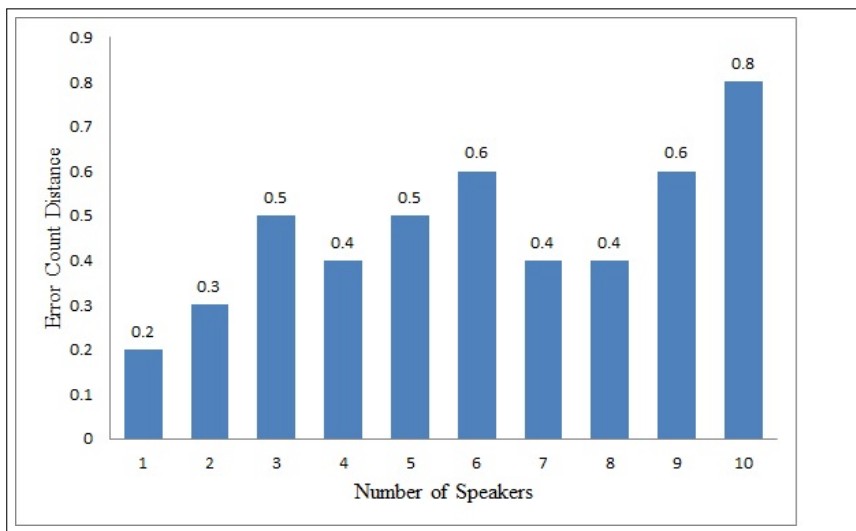

**Figure 8.** Error count distance single group scenario.

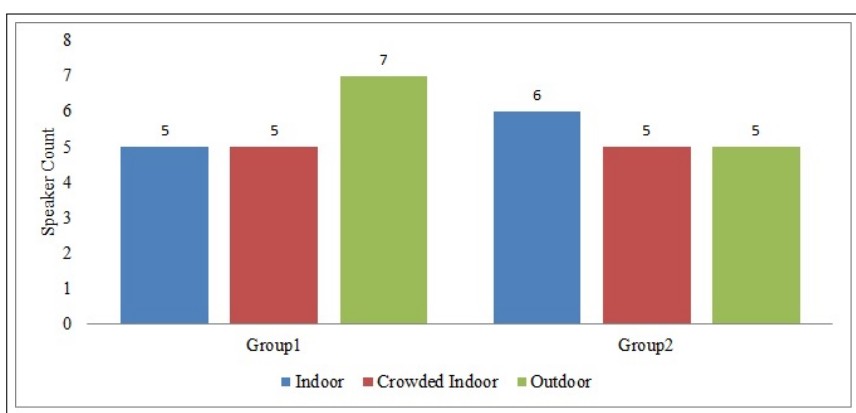

**Figure 9.** Speaker count at each SD in two-group scenario.

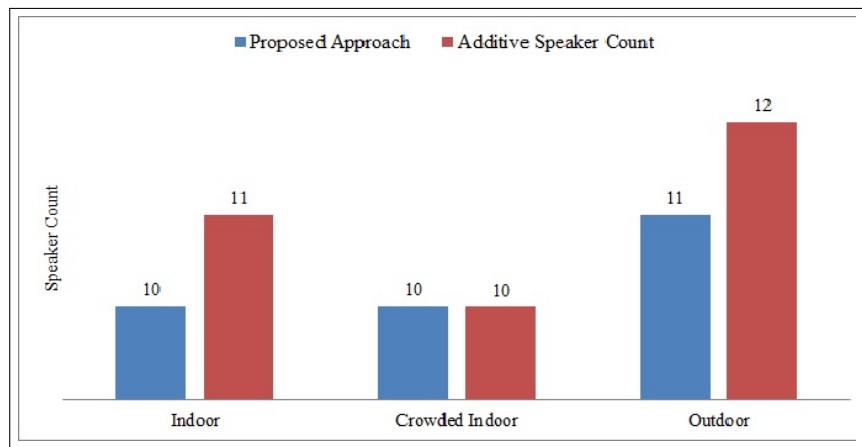

**Figure 10.** Comparing proposed methodology and additive speaker count strategy [5] for two-group scenario.

### 4.3.2. Performance of Three Group Scenario

We considered three speakers in the first two groups and four speakers in the third group in the three-group scenario. The other experimental setups were the same as above. Figure 11 shows the results of each SD. The figure shows that the indoor environment results were better than the other two cases due to low background noises. The declassification errors in the clustering of the statistical parameters were directly related to human emotion, health, environment, and various other factors. These variations could alter the speak count of a voice sample, as evident from Figure 6, affecting our accuracy in Figure 11.

Figure 12 shows a comparison between our distributed approach and additive approach of the speaker count for all three scenarios [5]. The average error count distance for all three environments was 0.755 for our methodology, which was better than the individual SD's summation results.

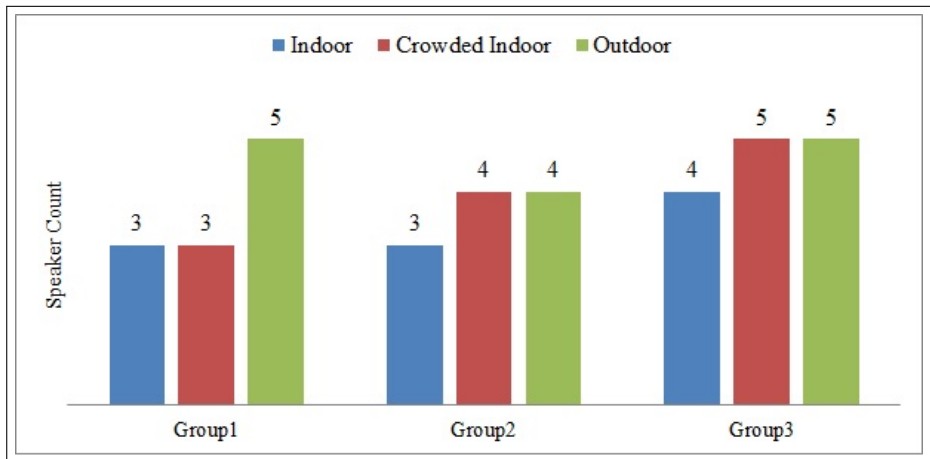

**Figure 11.** Speaker count at each SD in three-group scenario.

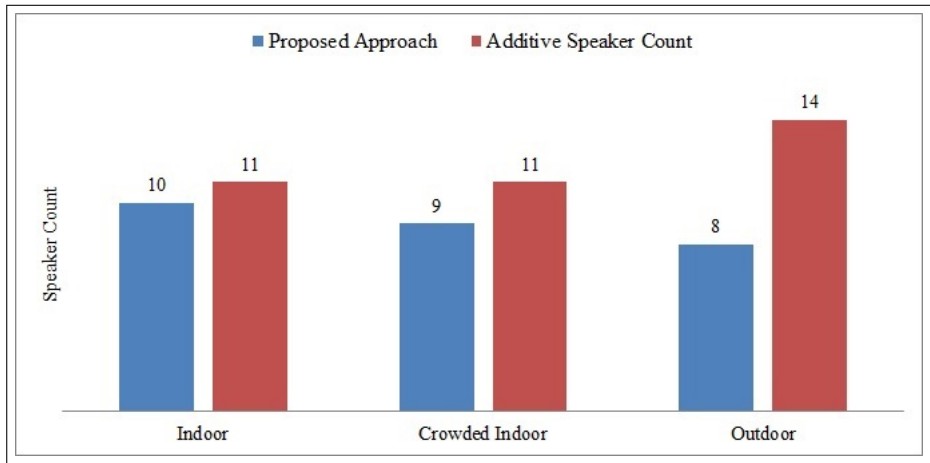

**Figure 12.** Comparing proposed methodology and additive speaker count strategy [5] for three-group scenario.

### 4.4. Performance in Indoor and Outdoor Environment

In the proposed solution, the speaker count was finally estimated by collating the exchanged codebook within a group to reduce errors due to the speaker's proximity to multiple microphones. Figure 13, shows the estimated speaker count for the three environments, as discussed above. The above discussed multi-group scenario derived the result. The figure shows that the indoor environments' results were 16% better than the outdoor environments, which could be due to the background noise.

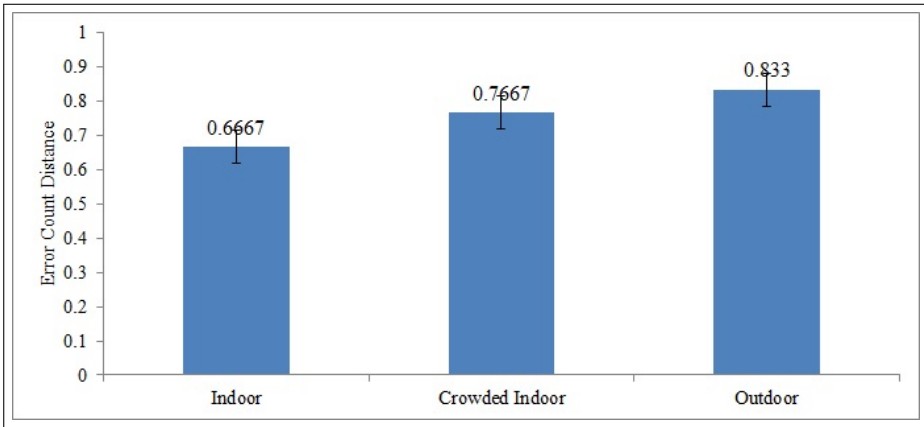

**Figure 13.** Average error count distance.

## 5. Discussion

In this section, we summarize the findings of the proposed solution to the distributed speaker count problem.

- Background Noise: The sound generated by TV or radio equipment can be a source of interference. However, the audio modulation techniques applied to the electronic medium (TV or radio) can be easily filtered out [35]. There are algorithms for source separation [36,37].
- Applications: The real-time distributed speaker count architecture can be used in a restaurant, movie theater, or shopping mall to rank the popularity of an event, object, or place. This is based on the assumption that a place's popularity is directly related to the number of people present nearby. Real-time ranking can help consumers make suitable choices. One can also use the methodology for determining audience participation in a lecture room for analyzing the popularity of lectures in the university.
- Complexity: The complexity of the *DataAcquisition()* and *FeatureExtraction()* modules discussed in Algorithm 2 is $O(n)$, where $n$ is the number of pitch samples collected. Similarly the complexity of the *FeatureClustering()* module $O(n^2)$ in worst case i.e., no two blocks merge, while on an average the complexity of *FeatureClustering()* module is $\theta(nlogn)$. Thus, the overall complexity of the proposed algorithm (i.e., Algorithm 1) in worst case is $O(n^2)$ and in average case $\theta(nlogn)$.
- Scalability: The proposed algorithm groups the SDs into disjoint clusters to increase the scalability of the system. Researchers have proposed various distributed approaches for clustering the nodes in a dynamic environment, which plays a role in handling a large number of nodes in a geographic area. The system's scalability can be further improved by using hierarchical clustering [22].
- Improvements: To improve the accuracy of the proposed methodology, we can include other statistical parameters, like median and average velocity of f0 change. Additionally, filtering the background noise can improve the accuracy in outdoor environments.

## 6. Conclusions

This paper proposes a distributed approach for the speaker count problem by clustering the statistical pitch parameters. In our model, we use multiple microphones that are readily available in today's smartphones to capture audio from an area of interest. The smartphones process the audio sample and extract the statistical pitch features. Finally, the features are shared with other smartphones to estimate the number of speakers in a neighborhood. The proposed technique can be useful for applications, such as real-time user ratings for movies or restaurants. We evaluate the algorithm performance with real implementation in a multigroup environment by capturing parallel conversations in both indoor and outdoor scenarios. The proposed distributed architecture reduces the speaker count error caused by the proximity of the microphones. In the future, we intend to ex-

tend our work for designing a real-time ranking system using the distributed speaker count architecture.

**Author Contributions:** Conceptualization, S.P. and A.B.; methodology, S.P.; software, S.P.; formal analysis, S.P.; investigation, S.P.; writing—original draft preparation, S.P.; writing—review and editing, A.B.; supervision, A.B. Both authors have read and agreed to the published version of the manuscript.

**Funding:** This research received no external funding.

**Institutional Review Board Statement:** Not applicable.

**Informed Consent Statement:** Not applicable.

**Data Availability Statement:** Data sharing is not applicable to this article.

**Conflicts of Interest:** The authors declare no conflict of interest.

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
