# Peer review of "A Distributed Approach to Speaker Count Problem in an Open-Set Scenario by Clustering Pitch Features"

_information, doi:10.3390/info12040157_

Round 1

Reviewer 1 Report

The authors propose an approach to estimate the speaker count problem, the proposal is based on working in a distributed system. In other words, the authors propose a system based on Smart Devices or smart phones, which are the elements that capture audio, the different capturing elements are grouped (manually) in groups (or cluster), so that one of the elements of the group performs controller tasks and communicates with other controllers. The main contribution of the article seems to be the task of the controllers to share information with other controllers, and thus reduce the errors caused by the capture of audio from the same speaker by two microphones of two different clusters. The technique used is the same as the one used to reduce the error due to the audio capture of the same speaker by two or more audio recorders within the same cluster. The management of the distributed system is not implemented, therefore the novelty is to test if the distributed system improves a system considered as a single unit in which there is a single controller that identifies speakers that are really the same and cause errors in the estimation of the number of speakers. And this is what should be improved in the article, by including scenarios with a greater number of clusters and speakers, and deciding the information sharing strategy between controllers, because if information is shared between all of them the system will have major drawbacks, related among other aspects such as “If we have two male or female speakers in an audio sample, there was a misclassification error of 10% in the clustering”, very important drawback for large scenarios.

On the other hand, the article needs significant improvement in other aspects:

  • Figures 3 and 5 should be flowcharts
  • Figure 6 must be made legible
  • The rest of the figures must also be legible.
  • It should be clear what the test scenario is and whether it has 20 speakers or 10 or 2 or 7.
  • Both the English language and numerous typos should be thoroughly reviewed, I do not list them here because there are too many (including bibliography)
  • Why is it not compared to other proposals? Explain more clearly.

Author Response

We thank the Editor and the Reviewers of MDPI Information for considering our paper and for giving us the opportunity to submit the revised manuscript. We believe that their valuable comments have helped us to improve our manuscript. We have tried our best to incorporate all suggestions and hope that it meets the standard of the Journal. In the following, we address the queries of the reviewers. Furthermore, we will be happy to incorporate further queries from the editor/reviewer on our paper.

Reviewer 2 Report

I have read with interest this paper, devoted to a future android application. The idea is interesting but the paper seems poor for me at the moment. The introduction presents the work without presenting some basic notions: the roots of the problem is not presented (inverse problem) with all its limitations and complexity. Then some important drawbacks are not discussed (noise treatment, echo cancellation, ...). The presented solution (clustering) is often affected by noise. My main drawback is about experiments. The authors only present a "one-speaker" result (fig.6)! It should be important to present 2 to 5 speakers clustering result... and the work 's hypotheis: Gaussian process (feature Pi)?

There are no performance indices: accuracy? separability?  sensibility to noise (TV set in a restaurant for instance),... In discussion the real-time applicability is not really discribed: complexity O(n) is not a real-time performance.

Thus I encourage the authors to furnish a more detailed version of their paper for publication.

Author Response

(The authors gave the same response as above.)

Round 2

Reviewer 1 Report

The article has been improved by the authors. 

Figure 6 should be explained in more detail to justify the conclusions drawn.

The article has not been sufficiently revised and still contains numerous typos, , both in the old and the new text.

Author Response

We thank the Editor and the Reviewers of MDPI Information for considering our paper and for giving us the opportunity to submit the revised manuscript. We believe that their valuable comments has helped us to improve our manuscript. We have tried our best to incorporate all suggestions and hope that it meets the standard of the Journal. In the following, we address the queries of the reviewers.

Reviewer 2 Report

The authors have adressed most of the remarks I made and I think the paper is now publishable.

Author Response

We also thank the respected reviewer for the critical assessment and valuable comments for improving the
manuscript. The reviewers’ comments have identified critical areas of the paper to improve overall presentation and clarity.

Round 3

Reviewer 1 Report

Please revise carefully. for example (just a sample):

modulation(PCM). (missing blank space)

Figure-6 (extra -)

algo-2

and some others

Author Response

We thank the Editor and the Reviewers of MDPI Information for considering our paper and for giving us the opportunity to submit the revised manuscript.
